# Arctigenin Attenuates Vascular Inflammation Induced by High Salt through TMEM16A/ESM1/VCAM-1 Pathway

**DOI:** 10.3390/biomedicines10112760

**Published:** 2022-10-31

**Authors:** Mengying Zeng, Ziyan Xie, Jiahao Zhang, Shicheng Li, Yanxiang Wu, Xiaowei Yan

**Affiliations:** 1Department of Cardiology, Peking Union Medical College Hospital, Chinese Academy of Medical Sciences & Peking Union Medical College, Beijing 100730, China; 2Key Laboratory of Endocrinology, Department of Endocrinology, Ministry of Health, Peking Union Medical College Hospital, Chinese Academy of Medical Sciences & Peking Union Medical College, Beijing 100730, China

**Keywords:** salt-sensitive hypertension, TMEM16A, ESM1, inflammation, vascular smooth muscle cells

## Abstract

Salt-sensitive hypertension is closely related to inflammation, but the mechanism is barely known. Transmembrane member 16A (TMEM16A) is the Ca^2+^-activated chloride channel in epithelial cells, smooth muscle cells, and sensory neurons. It can promote inflammatory responses by increasing proinflammatory cytokine release. Here, we identified a positive role of TMEM16A in vascular inflammation. The expression of TMEM16A was increased in high-salt-stimulated vascular smooth muscle cells (VSMCs), whereas inhibiting TMEM16A or silencing TMEM16A with small interfering RNA (siRNA) can abolish this effect in vitro or in vivo. Transcriptome analysis of VSMCs revealed some differential downstream genes of TMEM16A related to inflammation, such as endothelial cell-specific molecule 1 (ESM1) and CXC chemokine ligand 16 (CXCL16). Overexpression of TMEM16A in VSMCs was accompanied by high levels of ESM1, CXCL16, intercellular adhesion molecule-1 (ICAM-1), and vascular adhesion molecule-1 (VCAM-1). We treated VSMCs cultured with high salt and arctigenin (ARC), T16Ainh-A01 (T16), and TMEM16A siRNA (siTMEM16A), leading to greatly decreased ESM1, CXCL16, VCAM-1, and ICAM-1. Beyond that, silencing ESM1, the expression of VCAM-1 and ICAM-1, and CXCL16 was attenuated. In conclusion, our results outlined a signaling scheme that increased TMEM16 protein upregulated ESM1, which possibly activated the CXCL16 pathway and increased VCAM-1 and ICAM-1 expression, which drives VSMC inflammation. Beyond that, arctigenin, as a natural inhibitor of TMEM16A, can reduce the systolic blood pressure (SBP) of salt-sensitive hypertension mice and alleviate vascular inflammation.

## 1. Introduction

Hypertension, as a worldwide disease, is a major risk factor for cardiovascular disease, chronic kidney disease, and stroke. There is convincing evidence linking dietary salt intake and elevated blood pressure (BP) [1], and reduced sodium intake is an important approach to lowering BP [2]. Salt-sensitive hypertension is defined as a BP increased by 5% to 10% or at least 5 mmHg following an elevation in sodium intake [3]. However, individuals show different salt sensitivity for BP. Nearly 50% of the hypertensive and 25% of the normotensive population are affected [4]. The mechanisms that control how excess salt intake contributes to the elevation of BP remain unclear. Studies have shown the causes of salt-sensitive hypertension involve the kidney, vasculature, and central nervous system. Recently growing evidence suggests that inflammatory cells and their secreted effectors may contribute to hypertension and salt sensitivity [5]. On the one hand, elevated concentrations of sodium can activate T cells via the indirect activation of dendritic cells as well as direct activation of the salt-sensing kinase serum and glucocorticoid-regulated kinase 1 [6,7]. On the other hand, high salt intake has been shown to promote the production of isolevuglandin protein adducts (isoLG) that lead to activating T cells to raise BP [8]. These studies indicate there is a close relationship between salt sensitivity and inflammation, but the mechanisms are still unclear.

Vascular smooth muscle cells (VSMCs) can shift from a contractile to a synthetic phenotype in the event of hypertension [9]. The phenomenon is characterized by attenuated expression of contractile proteins along with an increased proliferation of VSMCs and functional enhancement of extracellular matrix (ECM) proteins, intercellular adhesion molecule (ICAM)-1 and vascular CAM (VCAM)-1, which can induce numerous white blood cells to infiltrate the vascular wall and trigger the inflammatory response, resulting in vascular damage and remodeling [10].

Transmembrane member 16A (Tmem16A), as a molecular regulator of the calcium-activated chloride channel, is primarily expressed in epithelial cells, smooth muscle cells, and sensory neurons [11], which can affect the contraction in vascular smooth muscle and BP in hypertensive rats [12]. A recent study suggested the involvement of TMEM16A in VSMC contraction and vascular remodeling for hypertension [13]. Furthermore, TMEM16A expression is closely related to the pathogenesis of infectious and non-infectious inflammatory diseases. It can promote inflammatory responses by increasing proinflammatory cytokine release [14]. Recently, studies found that arctigenin (ARC), as the major bioactive component of Fructus arctii in Chinese medicine, can inhibit TMEM16A both in vitro and in vivo [15,16] and is a novel TMEM16A natural product inhibitor. In addition, among plentiful inhibitors of TMEM16A, T16Ainh-A01 is the most potent and has the highest selectivity for TMEM16A [17]. However, little is known about the relationship between salt-sensitive hypertension and the role of TMEM16A in terms of vascular inflammation. Hence, our study provides a comprehensive insight into the effect of TMEM16A on the inflammation of vasculature in salt-sensitive hypertension.

## 2. Materials and Methods

### 2.1. Animals

All experimental procedures were performed with the approval and following the guidelines of the Peking Union Medical College Hospital Institutional Animal Care and Use Committee with approval letter Code (XHDW-2021-034,23-6-2021). Adult male C57B/L6J mice (8–10 weeks old, 19–22 g) were provided by the Chinese Academy of Medical Sciences. Mice were housed in standard conditions with a 12/12 h light/dark cycle and 19–22 °C room temperature with humidity in the range of 50 ± 10% and received a laboratory rodent diet. After 7 days of acclimation, mice were randomly divided into the normal salt (NS) group, L-NAME/high-salt (HS) group, HS+ arctigenin (50 mg·kg^−1^ body weight) (HS+ARC50) group, and HS+ arctigenin (100 mg·kg^−1^ body weight) (HS+ARC100) group. Mice in the HS group, HS+ARC50 group, and HS+ARC100 group were treated with NG-nitro-L-arginine methyl ester (L-NAME) (CAS No: 51298-62-5, Sigma-Aldrich, St. Louis, MO, USA), where L-NAME was dissolved in water at a concentration of 0.5 mg/mL for 2 weeks. After 2 weeks of washout, mice in the three groups were fed high-salt water (4%Nacl) to develop salt-sensitive hypertension. Next, after 6 weeks, mice in the HS+ARC50 group and HS+ARC100 group received treatment with arctigenin (CAS No:7770-78-7, MedChemExpress, Shanghai, China) (50 mg·kg^−1^ body weight) and arctigenin (100 mg·kg^−1^ body weight) once daily for consecutive 4 weeks, respectively (Figure 1).

### 2.2. Blood Pressure Measurement

A non-invasive tail-cuff device (Volume Pressure Recording, CODA, Kent Scientific Corp, Torrington, CT, USA) was used to measure and record systolic blood pressure (SBP), diastolic blood pressure, and heart rate in conscious mice weekly. Before the measurement, animals were placed in a specialized holder for 10–15 min to acclimate to the surroundings. At the end of the experimental period, mice were euthanized by intraperitoneal injection of a lethal dose of pentobarbital, then thoracic aortas were excised and fixed in 4% paraformaldehyde for studies.

### 2.3. H&E Staining and Slide Analysis

The aortic samples (NS, HS, HS+ARC50, and HS+ARC100 groups) were infiltrated into 4% paraformaldehyde for 24 h to dehydrate and fix the tissues. Subsequently, we trimmed and embedded the tissues in paraffin. They were sectioned (4 μm sections) and stained with hematoxylin and eosin (HE). Next, aortic intima-media thickness was quantitated in these groups, and an optical microscope (Japan Nikon, NIKON ECLIPSE CI) was used to analyze these slides. For each aorta, the thickness of the media layer was measured from the endothelial surface to the adventitia in 10 fields, using the CaseViewer, 3DHISTECH, version 2.4.0.

### 2.4. Vascular Smooth Muscle Cells

Mouse aortic vascular smooth muscle cells (MAo, VSMCs) were purchased from Procell (Cat NO: CP-M076, Procell Life Science & Technology Co., Ltd., Wuhan, China), and 5 × 10^5^ VSMCs/cm^2^ were grown in smooth muscle growth media (Cat NO: #1101, ScienCell Research, San Diego, CA, USA) supplemented with 10 mL fetal bovine serum (FBS) (Cat NO: #0010, ScienCell Research, San Diego, CA, USA) to 2% final concentration, 5 mL of smooth muscle cell growth supplement (Cat NO: #11152, ScienCell Research, San Diego, CA, USA) and 5 mL of penicillin/streptomycin solution (Cat NO: #0503, ScienCell Research, San Diego, CA, USA). All cells were grown at 37 °C in a 5% CO_2_ incubator and used from passages 2–4.

### 2.5. RNA-Sequencing (RNA-Seq) Analysis

VSMCs (5 × 10^5^/cm^2^) were seeded onto 6-well plates and divided into the normal salt (NS) group (Nacl: 118 mmol/L), high salt (HS) group (Nacl: 169 mmol/L), HS+ARC group (arctigenin: 100 μM, CAS No: 7770-78-7, MedChemExpress, Shanghai, China) and HS+ T16Ainh-A01 (HS+T16) group (T16Ainh-A01: 10 μM, CAS No: 552309-42-9, MedChemExpress, Shanghai, China). All the cells were cultured in smooth muscle growth media, and we changed the growth media every two days. Cells in the HS, HS+ARC, and HS+T16 groups were cultivated by growth media with a solution containing 169 mmol/L sodium. After 48 h of culture with high sodium media, cells in the HS+ARC group were subjected to ARC (100 μM) intervention for 24 h, and cells in the HS+T16 group received T16 (10 μM) intervention for 30 min. The total RNA of cells in all groups was extracted using Trizol regent (CAT No: 15496018, Invitrogen, Carlsbad, CA, USA). Then, RNA sequencing was performed, and its data were analyzed by Novogene (Beijing, China).

### 2.6. Immunofluorescence

According to the results of RNA-sequencing analysis, the inflammation- and TMEM16A-related differentially expressed gene was chosen for further validation by immunofluorescence in aortic paraffin sections (NS, HS, and HS+ARC100 groups), described above. The slides were deparaffinized by washing with xylene I, xylene II, and absolute ethyl alcohol each for 15 min; absolute ethyl alcohol, 85% ethanol, and 75% ethanol each for 5 min and distilled water successively in a decolorization shaker (Beijing 61 Biological Technology Co., Ltd., Beijing, China). These sections were placed in citrate solution (PH: 6.0, CAT No: P0026, Pinuofei Biological Technology Co., Ltd., Wuhan, China) for antigen retrieval at 37 °C for 10 min, rinsed 3 times with PBS (PH: 7.4) for 5 min/time in the shaker and incubated overnight at 4 °C with monoclonal rabbit anti-TMEM16A (ab64085, Abcam, 1:100, Shanghai, China), polyclonal rabbit anti-endothelial cell-specific molecule 1 (ESM1) (PA5-47237, Invitrogen, 1:200, Carlsbad, CA, USA), monoclonal rabbit anti-ICAM (ab222736, Abcam,1:50, Shanghai, China) and monoclonal rabbit anti-VCAM (ab134047, Abcam, 1:200, Shanghai, China). Next, after rinsing 3 times with PBS for 5 min/time in the shaker, the sections were then incubated for 50 min at room temperature with goat anti-rabbit (115-165-003, Jackson 1:200, Lancaste, PA, USA) and donkey anti-goat (GB22404, Google, 1:200) and covered with DAPI (C0060, Solarbio, 1:100, Beijing, China) lucifugal for 10 min. Lastly, images were captured using a fluorescence microscope (Japan Nikon, NIKON ECLIPESE TI-SR). Immunofluorescence was followed by Image J (Fiji software version 2.0) quantitative analysis.

### 2.7. TMEM16A and ESM1 Silencing

Small interfering RNA targeting the TMEM16A gene (siTEME16A) was purchased from OriGene (CAT No: SR411408, OriGene China, Wuxi, China). Similarly, ESM1 expression was inhibited using specific ESM1 siRNA (siEMS1) (RIBO, Guangzhou, China). VSMCs were seeded into 6-well cell culture plates and incubated for 48 h. We used a riboFECT CP Transfection Kit (CAT No: C10511-05, RIBO, Guangzhou, China) and riboFECT CP buffer (CAT No: C10502-05, RIBO, Guangzhou, China) to perform transfection. Then, cells were subjected to transfection with siTMEM16A and siESM1 for 24 and 48 h, respectively. Additionally, the transfection efficiency of TMEM16A and ESM1 silencing was evaluated at the protein level with Western blot analysis.

### 2.8. Western Blot Analysis

Proteins of VSMCs were extracted with radioimmunoprecipitation assay (RIPA) buffer (CAT No: R0020, Solarbio, Beijing, China) and quantified using the Pierce BCA Protein assay kit (CAT No: 23227, Invitrogen, Carlsbad, CA, USA). The optical density was measured at 562 nm with a microplate reader (Synergy H1, BioTek, Winooski, VT, USA). Subsequently, proteins were separated by 10% sodium dodecyl sulfate–polyacrylamide gel electrophoresis (SDS–PAGE) and transferred to 0.45 μm polyvinylidene fluoride (PVDF) membranes (CAT No: IPVH00010, MedChemExpress, Shanghai, China). The membranes were then incubated overnight at 4 °C with primary antibodies, including TMEM16A recombinant rabbit monoclonal antibody (ab64086, Abcam, 1:1000, Shanghai, China), vascular adhesion molecule-1 (VCAM-1) recombinant rabbit monoclonal antibody (ab134047, Abcam, 1:2000), intercellular adhesion molecule-1 (ICAM-1) recombinant rabbit monoclonal antibody (ab222736, Abcam, 1:1000, Shanghai, China), GAPDH recombinant rabbit monoclonal antibody (ab181602, Abcam, 1:10,000, Shanghai, China) and ESM1 goat polyclonal antibody (PA5-47237, Invitrogen, 1:2000, Carlsbad, CA, USA). The membranes were incubated with secondary IgG antibodies (ab6728, Abcam, 1:2000; ZB-2306, ZSGB-BIO, 1:2500, Shanghai, China) after washing for 1 h at room temperature. HRP activity was detected with a chemiluminescent substrate (CAT No: WBKLS0100, MedChemExpress, Shanghai, China). Quantitative analysis of protein expression was carried out with Image J (Fiji software version 2.0).

### 2.9. Quantitative Enzyme-Linked Immunosorbent Assay (ELISA)

VSMC culture supernatants were collected in the NS, HS, T16+HS, ARC+HS, siTMEM16A+HS and siESM1 +HS groups, then cytokine production was quantified using a CXC chemokine ligand 16 (CXCL16) mouse ELISA kit (1.3 pg/mL, 1.25–80 ng/mL, CAT No: EMCXCL16, Invitrogen, Carlsbad, CA, USA). We used the protocols provided by the manufacturer.

### 2.10. Macrophage–VSMC Adhesion Assay

The adhesion assay was performed in a 24-well plate coated with 4% agarose. Macrophages were resuspended and incubated with 5 μM Calcein-AM (CAT No: C8950, Solarbio, Beijing, China) for 30 min at 37 °C. A total of 5 × 10^5^ labeled macrophages were added into the 24-well plate with prepared VSMCs in the NS, HS, T16+HS, ARC+HS, siTMEM16A+HS, and siESM1 +HS groups and were allowed to adhere for 1 h at 37 °C in 5% CO_2_. The non-adherent cells were removed by PBS, and the fluorescence intensity of residual macrophages was detected via fluorescence microscopy (OLYMPUS, IX71, Tokyo, Japan).

### 2.11. Statistical Analysis

Statistical data analysis was performed using the GraphPad Prism version 8 (GraphPad Software, San Diego, CA, USA) software. Kolmogorov–Smirnov test is used to check the normality of the distribution of the results. According to the type of data, the statistical significance was determined by using the Chi-square and t-test between the two groups. Differences among three or more groups were determined using one-way analysis of variance (ANOVA) and non-parametric ANOVA tests, and post hoc analysis with Sidak correction was applied All *p*-values were two sides, and *p* < 0.05 was considered statistically significant.

## 3. Results

### 3.1. L-NAME/High-Salt-Induced Hypertension and Systolic Blood Pressure Reduction after Arctigenin Intervention

After treatment with L-NAME/high salt, the SBP of mice was significantly increased compared with that in the NS group (*p* < 0.0001). After 4 weeks of ARC treatment, there was a significant decline in SBP in the HS+ARC100 group (*p* < 0.0001), but a significant SBP difference between the HS group and the HS+ARC50 group was not observed (Figure 2A). Meanwhile, H&E staining revealed that aortic intima-media was significantly thickened in the HS group compared to that in the NS group (*p* < 0.0001), and arctigenin was capable of decreasing the thickness induced by L-NAME/high salt (*p* < 0.0001) (Figure 2B). These results suggested that a salt-sensitive hypertension mouse model was established. Moreover, arctigenin had a potential antihypertensive effect.

### 3.2. Increased Expression of TMEM16A in Aortic VSMCs in High-Salt Environment

Western blot analysis showed increased TMEM16A expression of VSMCs cultured with high-salt media, as shown in Figure 3A,B (*p* < 0.05). We used immunofluorescence experiments to confirm the upregulation of TMEM16A protein in high-salt-cultured aortic VSMCs. The results showed TMEM16A was expressed in VSMCs, and the fluorescence intensity of TMEM16A in the HS group was significantly increased compared with that in the NS group (Figure 3C,D) Moreover, after arctigenin treatment (100 mg/kg), obvious decreased expression of TMEM16A could be observed in the aortic VSMCs compared with HS group (Figure 3D,E).

### 3.3. Targeted Inhibition of TMEM16A Attenuated VCAM-1 and ICAM-1 Expression In Vitro and In Vivo

To clarify the role of TMEM16A in inflammation induced by high-salt, we used aortic VSMCs cultured with ARC, T16, and siTMEM16A to perform Western blotting. Firstly, results showed a discernible reduction in TMEM16A proteins in VSMCs after treating with 50, 100, and 200 μM arctigenin for 24 h (Figure 4A). Then, 100 µM arctigenin was used to perform the verification. The 30 µM siTMEM16 knockdown TMEM16A protein for 24 h in VSMCs also was confirmed by Western blotting (Figure 4B). Western blotting analysis showed that the expression of VCAM-1 and ICAM-1, closely related to inflammation, was significantly decreased after 100 µM arctigenin for 24 h, 10 µM T16Ainh-A01 for 30 min, and 30 µM siTMEM16A for 24 h (Figure 4C). In addition, to test the relationship of TMEM16A with VCAM-1 and ICAM-1, immunostaining was performed in mouse aortic VSMCs and showed increased expression of VCAM-1 and ICAM-1 in the HS group. However, obvious decreased expression of VCAM-1 and ICAM-1 was observed when TMEM16 was inhibited by arctigenin (Figure 4D).

### 3.4. High-Salt-Induced Differentially Expressed Downstream Genes of TMEM16A in Aortic VSMCs

To investigate the role of TMEM16A in high-salt-induced inflammation, quantitative analysis of gene expression was performed using RNA extracted from groups of aortic VSMCs. The mRNA expression level and abundance were calculated as FPKM [18] (Figure 5A). Clustering analysis was used to describe the distribution of differential expression genes in every sample; genes with similar expression patterns in the clustering heatmap were clustered together and those with increased expression values were coded from green to red (Figure 5B). Transcript analysis revealed a broad spectrum of differential expression genes between the two groups. The overall distributions of these genes are represented in the volcano plots in Figure 5C–E. Adjusted p-values (Padj < 0.05) and logFC > 1 were set as the threshold. Venn diagrams showed the overlap of different genes between multiple sets; four genes were detected in the first dataset, including ucp1, Esm1, cyp26b1, and Fos (Figure 5F). On the contrary, 24 common genes were identified in the dataset of the NS and HS group versus the HS and T16 group, such as Esm1, Fos, Ucp1, Ccl2, and Cxcl16 (Figure 5G). To further investigate cell functional states and potential molecular regulators, Gene Ontology (GO) and Kyoto Encyclopedia of Genes and Genomes (KEGG) analyses were used to perform functional classification of these downregulated genes in the HS+ARC/HS+T16 groups compared to the HS group in Aortic VSMCs. The GO enrichment bar charts (Appendix A) revealed the biological process, cellular components and molecular function in the GO enrichment of differential expressed downstream genes of TMEM16A. Both groups showed a higher proportion of biological process genes, including positive regulation of cell migration, positive regulation of cell motility, and positive regulation of cellular components. At the same time, the items with the most significant activity in cellular components were focal adhesion, cell-substrate adherens junction, and anchoring junction in the group of HS and ARC. On the other hand, the group of HS and T16 revealed some significant activity of cellular components, such as regulation of axonogenesis, the adherens junction, and the cell–cell junction. Furthermore, some items were significantly enriched in molecular function in the group of HS and ARC, such as phosphoric ester hydrolase activity, chemorepellent activity, and 3′,5′-cyclic-AMP phosphodiesterase activity. Most items in the group of HS and T16 were significantly enriched in chemorepellent activity, nucleoside–triphosphatase regulator activity, and enzyme activator activity. Beyond that, there were 20 prominent KEGG pathways in the HS and ARC group and HS and T16 group. As shown in Appendix A, differentially expressed genes in aortic VSMCs between HS and ARC and HS and T16 groups were significantly enriched in parathyroid hormone synthesis, secretion, and action: MicroRNAs in cancer and proteoglycans in cancer and MicroRNAs in cancer and ECM–receptor interaction.

### 3.5. TMEM16A Inhibition Largely Attenuates ESM1-Stimulated CXCL16/VCAM-1 Pathway

As mentioned above, ESM1 and CXCL16 were determined to be differential genes downstream of TMEM16A through differential gene analysis. ESM1 is considered a secreted protein that plays a role in inflammation and angiogenesis [19,20]. CXCL16 is a chemotactic cytokine that affects the recruitment of T cells to sites of inflammation [21]. It is essential for increasing vascular inflammation via migration and recruitment of IL-17-producing T cells [22]. We inferred that ESM1 and CXCL16 might be involved in regulating VSMC inflammation. We began by studying the ESM1 expression under different conditions. Western blotting and immunostaining analysis showed that the expression of ESM1 was significantly increased in VSMCs with HS; in contrast, TMEM16A inhibition diminished the levels of ESM1 (Figure 6A,B). Furthermore, we assessed the effect of ESM1 on VCAM-1 and ICAM-1 expression in aortic VSMCs, in which ESM1 was knocked down by siRNA ESM1 (siESM1). Primarily, data in Figure 6C confirmed that siESM1 (30 μM, 48 h) was effective in reducing target protein expression. Next, ESM1-specific silencing largely decreased VCAM-1 and ICAM-1 protein expression (Figure 6D). Lastly, the culture supernatant levels of CXCL16 in every group were detected by ELISA, and a significantly higher level was found in the HS group compared to the NS group, but after treatment with T16 for 30 min, ARC for 24 h, siTMEM16A for 24 h and siESM1 for 48 h, a significant decline in CXCL16 was observed (Figure 6E).

### 3.6. TMEM16A Deficiency Led to Reversed Macrophage Adhesion in VSMCs Induced by High Salt

To clarify the role of TMEM16A in VSMC function in vitro, the macrophage adhesion assay was performed. Results demonstrated that adherent fluorescent macrophages were significantly increased in the HS group compared to the NS group. The adhesion between macrophages and high-salt-treated VSMCs was suppressed by VSMC TMEM16A deficiency or ESM1 silencing (Appendix A).

## 4. Discussion

Salt-sensitive hypertension is closely related to inflammation, and there are convincing studies demonstrating that a high-salt diet can polarize immune cells towards an inflammatory phenotype, but the specific mechanisms are still unclear. In this work, we revealed the crucial role of TMEM16A for aortic vascular inflammation in an LNAME/high-salt-induced hypertension mouse model. Firstly, we demonstrated the significantly higher expression of TMEM16A in aortic vascular smooth muscle both in vivo and in vitro. Moreover, arctigenin can significantly inhibit TMEM16 expression both in vivo and in vitro and reduce the SBP of mice with salt-sensitive hypertension. In addition, TMEM16A regulates the expression of the matricellular protein (ESM1) and reduces TMEM16A activity, effectively attenuating adhesion molecules (VCAM-1 and ICAM-1) expression. Ultimately, the inhibition effect of arctigenin can be translated into an anti-inflammatory in an LNAME/high-salt-induced salt-sensitive hypertension mouse model.

Increasing evidence indicates that TMEM16A dysfunction contributes to hypertension. TMEM16A can regulate endothelial reactive oxygen species generation via Nox2-containing NAPDH oxidase; hence, endothelial dysfunction and hypertension develop [23]. In the present study, it was determined that inhibition of TMEM16A remarkably reduced the SBP of mice with salt-sensitive hypertension, which means TMEM16A may be a crucial protein in the development of salt-sensitive hypertension. Previous studies revealed that TMEM16A played an important role in promoting inflammatory cytokine release from tissue cells [24,25]. TMEM16A overexpression also contributed to increased vascular permeability and leukocyte recruitment in endothelial cells [26]. It is noteworthy that inflammation may play a dominant pathological role in TMEM16A-associated diseases, whereas its contribution to VSMC inflammation has been barely explored. In our in vivo and in vitro studies, after the inhibition of TMEM16A with ARC, T16, or siTMEM16A, VCAM-1 and ICAM-1 expressions were blocked in TMEM16A overexpressed VSMCs. Some studies indicated that adhesion molecules, notably VCAM-1 and ICAM-1, were direct targets of miRNA in response to inflammatory stimuli [27]. Upregulation of VCAM-1 and ICAM-1 facilitated neutrophil adhesion and infiltration, which were considered pro-inflammatory chemokines. These results indicated that TMEM16A is predominantly responsible for high-salt-stimulated VSMC inflammation induced by high salt in vivo and in vitro by regulating VCAM-1 and ICAM-1. Hence, the protective effect of arctigenin against vascular inflammation in vivo was likely mediated by the inhibition of adhesion molecules. Arctigenin was first confirmed as a TMEM16A inhibitor for lung adenocarcinoma [16]. It is the main active ingredient of burdock that can act as an anti-inflammatory agent in traditional Chinese medicine. One previous study indicated that arctigenin had antihypertensive effects in spontaneously hypertensive rats [28]. Taken together, our results and literature evidence unequivocally support the finding that arctigenin played a prominent antihypertensive and anti-inflammatory role via inhibiting TMEM16A.

Verifying TMEM16A was predominantly responsible for high-salt-stimulated VSMC inflammation, which enabled us to further investigate its regulator mechanisms. Here, ESM1 and CXCL16 were confirmed as downstream genes of TMEM16A via RNA-Seq analysis. ESM1 was initially identified as an endothelial cell-specific marker mainly produced by endothelial cells. As mentioned above, TMEM16A can influence the function of endothelial cells. However, to the best of our knowledge, there is no report on TMEM16A-regulated expression of matricellular proteins. This regulation was identified by inducible TMEM16A inhibiting or silencing in vivo or in vitro. Moreover, silencing ESM1 maintained inactive VCAM-1 and ICAM-1 in in vitro studies, which means TMEM16A expression may increase vascular inflammation by regulating the ESM1/VCAM-1 pathway. This finding is significant; on the one hand, TMEM16A has the potential to be further investigated as a key protein in vascular inflammation for salt-sensitive hypertension. On the other hand, particularly given that the majority of studies on ESM1 have been focused on endothelial cells, the expression and functional regulation of ESM1 in VSMCs are little known. For example, ESM1 plays a role in inflammation and is involved in endothelial inflammation [29]. It can inhibit the adhesion of leukocytes to ICAM1, which suggests a negative role in leukocyte extravasation during inflammation [30]. In this regard, our work has led to another finding concerning ESM1’s function in prompting VSMC inflammation.

CXCL16 is one of two known membrane-anchored chemokines [31,32] that can activate endothelial cells to upregulate ICAM-1 [33]. We performed ELISA of culture supernatant samples obtained in multiple groups, and there were significant positive correlations between the levels of CXCL16 and the levels of TMEM16A and ESM1. It is noteworthy that the downregulation of TMEM16A or ESM1 expression was accompanied by reduced CXCL16 expression, which hints that TMEM16 and ESM1 may affect VCAM-1 and ICAM-1 expression via regulating CXCL16.

In summary, our results outlined a signaling scheme in which increased TMEM16 protein/activity upregulated ESM1, which possibly activated the CXCL16 pathway and increased VCAM-1 and ICAM-1 expression, thus driving VSMC inflammation. Beyond that, arctigenin, as a natural inhibitor of TMEM16A, can reduce SBP of salt-sensitive hypertension mice and alleviate vascular inflammation.

## Figures and Tables

**Figure 1 biomedicines-10-02760-f001:**
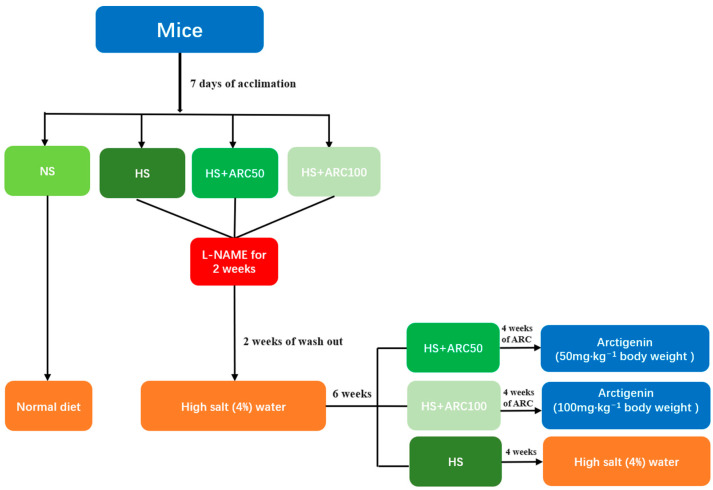
Flow chart of animal model.

**Figure 2 biomedicines-10-02760-f002:**
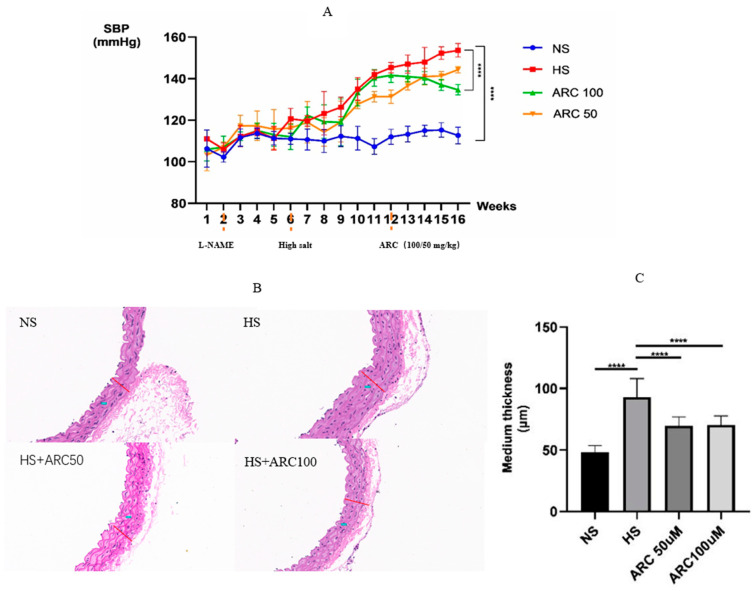
(**A**) SBP in different groups from the first week to the 16th week. Mice in the HS, HS+ARC50, and HS+ARC100 groups received L-NAME in the second week for 2 weeks and were fed with high-salt water from the sixth week. Mice received ARC treatment for 4 weeks after L-NAME/high-salt diet. SBP between NS and HS, HS and HS+ARC100 groups showed significant difference using one-way ANOVA and *p*-value adjusted for multiple comparisons with Sidak multiple-comparison test (*n* = 4), **** *p* < 0.0001. (**B**) Aortic intima-media thickness in different groups, (**C**) Quantitative analysis of aortic intima-media thickness in every group, showing the significant difference between NS and HS groups, HS group and ARC 50/ARC 100 groups using one-way ANOVA and P-value adjusted for multiple comparisons with Sidak multiple-comparison test (*n* = 4), **** *p* < 0.0001. NS: normal salt diet group, HS: L-NAME/high-salt group, HS+ARC50: L-NAME/high salt + arctigenin (50 mg/kg) group, HS+ARC100: L-NAME/high salt + arctigenin (100 mg/kg) group.

**Figure 3 biomedicines-10-02760-f003:**
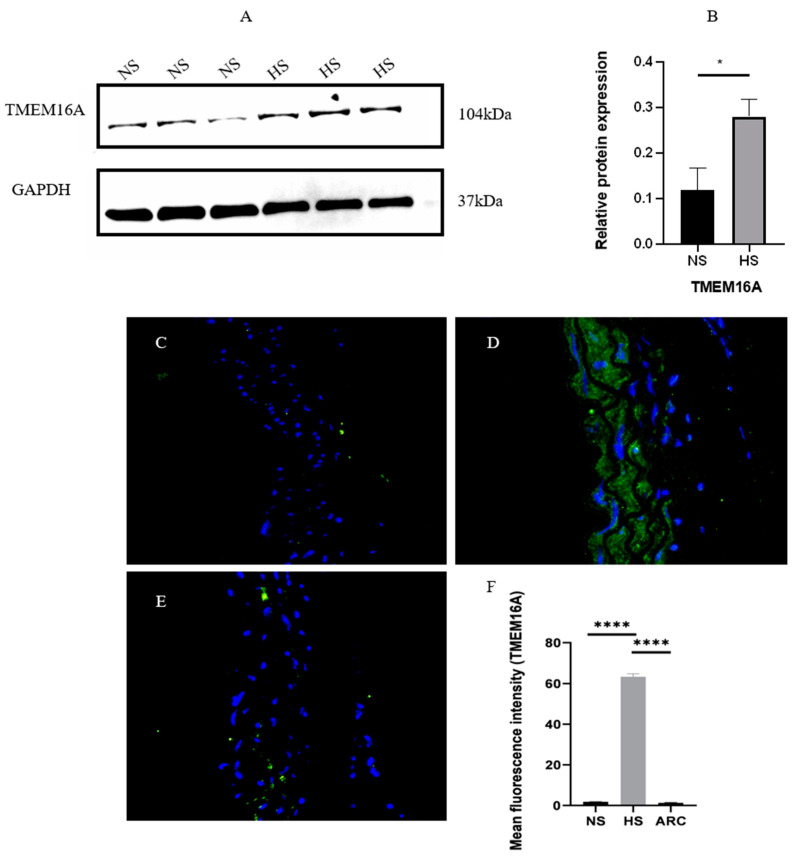
(**A**) Western blot analysis indicated increased expression of TMEM16A in the HS group compared with NS group. (**B**) Relative level of TMEM16A protein expression. Data are expressed as mean ± SD, * *p* < 0.05, *t*-test (*n* = 3). (**C**–**E**) Representative images of TMEM16A (green) immunofluorescence in thoracic aortas in the NS group, HS group, and HS+ARC100 groups (*n* = 4). Scale bar: 50 µm; blue represents nucleus. (**C**) NS group; (**D**) HS group; (**E**) HS+ARC100 group. (**F**) Mean fluorescence was quantified using one-way ANOVA and *p*-value adjusted for multiple comparisons with Sidak multiple-comparison test (*n* = 4), **** *p* < 0.0001. NS: normal salt group, HS: high-salt group, HS+ARC: HS+ arctigenin (100 mg/kg) group.

**Figure 4 biomedicines-10-02760-f004:**
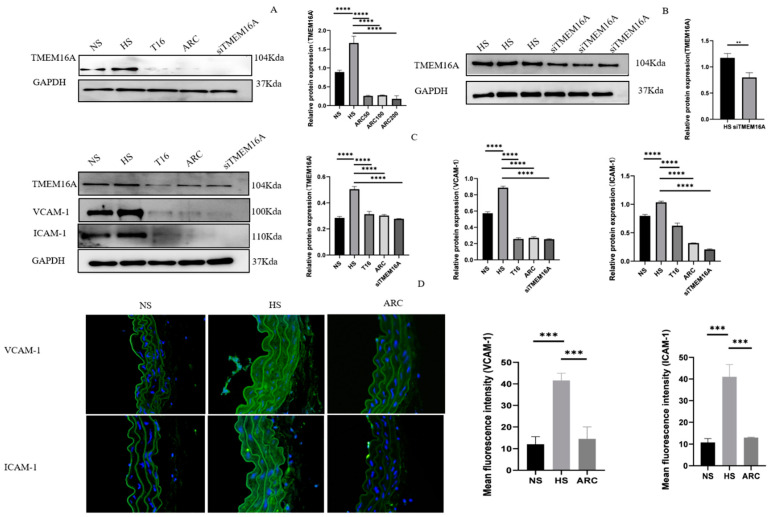
(**A**) Western blotting results showed discernible reduction in TMEM16A proteins in VSMCs after treating with 50, 100, and 200 μM arctigenin for 24 h, mean protein was quantified using one-way ANOVA and *p*-value adjusted for multiple comparisons with Sidak multiple-comparison test (*n* = 3), **** *p* < 0.0001; (**B**) 30 μM siTMEM16A knockdown TMEM16A protein for 24 h in the TMEM16A knockdown cells compared to the HS group. Data are expressed as mean ± SD, ** *p* < 0.01 *t*-test (*n* = 3); (**C**) The Western blotting results showed that incubating aortic VSMCs with 10 μM T16Ainh-A01, 100 μM arctigenin and 30 μM siRNA TMEM16A led to a decrease in TMEM16A, VCAM-1 and ICAM-1. Data are expressed as mean ± SD, and relative protein expression was quantified using one-way ANOVA and *p*-value adjusted for multiple comparisons with Sidak multiple-comparison test (*n* = 3), **** *p* < 0.0001. (**D**) Microscopic images showed immunofluorescence staining of VCAM-1 and ICAM-1 (green) in thoracic aortas (*n* = 4); blue represents nucleus. Mean fluorescence was quantified using one-way ANOVA and *p*-value adjusted for multiple comparisons with Sidak multiple-comparison test (*n* = 4), *** *p* < 0.001. NS: normal salt group, HS: high-salt group, T16: HS+ T16Ainh-A01 group, ARC: HS+ arctigenin (100 mg/kg) group, siTMEM16A: HS+siRNA TMEM16A group. siEMS1: HS+ siRNA ESM1 group.

**Figure 5 biomedicines-10-02760-f005:**
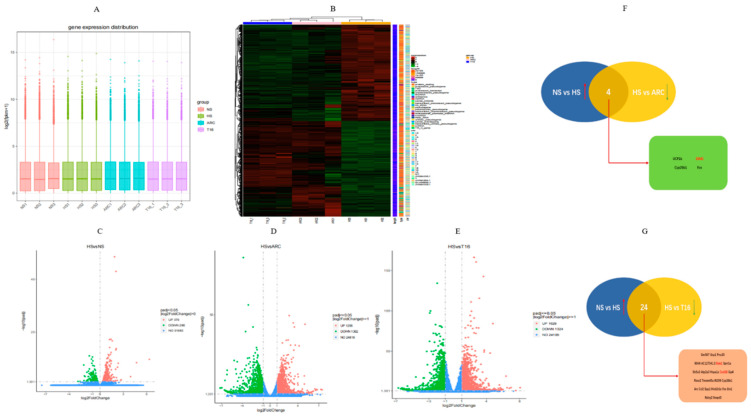
(**A**) FPKM box plot of the expression levels of transcripts in each group. (**B**) Heatmap clustering analyses of differential expression genes in each sample; obvious green color represents low gene expression, while red represents high expression. (**C**–**E**) Volcano plots showing differential expression genes between each group with different increased genes colored in red decreased genes colored in green and non-differential genes colored in blue. (**F**,**G**) Common differential expression genes are identified in Venn diagrams between these upregulated genes with logFC > 1 in the HS group compared with NS group and all downregulated genes in the ARC/T16 groups compared with HS group. NS: normal salt group, HS: high-salt group, ARC: HS+ arctigenin (100 mg/kg) group, T16: HS+ T16Ainh-A01 group.

**Figure 6 biomedicines-10-02760-f006:**
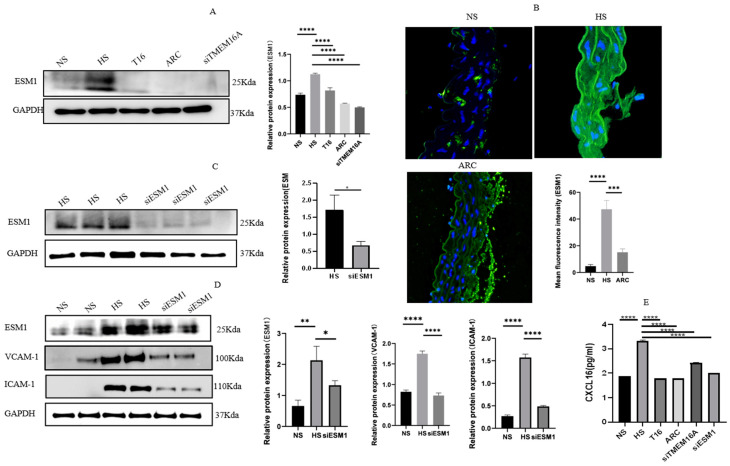
(**A**) The Western blotting results showed that incubating aortic VSMCs with 10 μM T16Ainh-A01, 100 μM arctigenin, and 30 μM siRNA TMEM16A led to a decrease in ESM1. Data are expressed as mean ± SD, and relative protein expression was quantified using one-way ANOVA and *p*-value adjusted for multiple comparisons with Sidak multiple-comparison test (*n* = 3), **** *p* < 0.0001. (**B**) Microscopic images showed immunofluorescence staining of ESM1 (green) in thoracic aortas (*n* = 4); blue represents nucleus. Mean fluorescence was quantified using one-way ANOVA and *p*-value adjusted for multiple comparisons with Sidak multiple-comparison test (*n* = 4), *** *p* < 0.001, **** *p* < 0.0001. (**C**) 30uM siESM1 knockdown EMS1 protein for 48 h in the ESM1 knockdown cells compared to the HS group. Data are expressed as mean ± SD, * *p* < 0.05, *t*-test (*n* = 3). (**D**) ESM1, VCAM-1, and ICAM-1 protein expression dramatically declined after silencing ESM1 in VSMCs. Data are expressed as mean ± SD, and relative protein expression was quantified using one-way ANOVA and *p*-value adjusted for multiple comparison with Sidak multiple-comparison test (*n* = 3), * *p* < 0.05, ** *p* < 0.01, **** *p* < 0.0001. (**E**) Aortic VSMCs were cultured with NS, HS, ARC, siTMEM16A siESM1 for 24 h and T16 for 30 min, and the levels of CXCL16 in the culture media were quantified using the enzyme-linked immunosorbent assay (ELISA). Data are expressed as mean ± SD, and relative CXCL16 expression was quantified using one-way ANOVA and *p*-value adjusted for multiple comparisons with Sidak multiple-comparison test (*n* = 3), **** *p* < 0.0001 (*n* = 3). NS: normal salt group, HS: high-salt group, T16: HS+ T16Ainh-A01 group, ARC: HS+ arctigenin (100 mg/kg) group, siTMEM16A: HS+siRNA TMEM16A group. siEMS1: HS+ siRNA ESM1 group.

## Data Availability

Not applicable.

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
