# Peer review of "Arctigenin Attenuates Vascular Inflammation Induced by High Salt through TMEM16A/ESM1/VCAM-1 Pathway"

_biomedicines, 2022, doi:10.3390/biomedicines10112760_

Round 1

Reviewer 1 Report

In the present manuscript, the authors investigate the role of Arcrigenin on the vascular inflammation induced by salt-sensitive hypertension. The authors show that TMEM16A, a Ca2+-activated chloride channel, expressed in smooth muscle cells, can promote inflammatory responses by increasing proinflammatory cytokine release. In fact, in high-salt group the TMEM16A expression levels were increased compared with normal salt group in vivo. The expression of TMEM16A was increased in high-salt-stimulated vascular smooth muscle cells (VSMCs), whereas inhibiting TMEM16A or silencing TMEM16A with small interfering RNA (siRNA) can abolish this effect in vitro. In conclusion, they demonstrate that TMEM16 protein upregulated ESM1, which possibly activated the CXCL16 pathway and increased VCAM-1 and ICAM-1 expression, which drives VSMC inflammation.

Although the manuscript is interesting, some issues should be reconsidered before publication. Listed below are some specific comments.

-In the abstract there is a small mistake: the abbreviation EMS1 instead of ESM1

- Since the normotensive population show salt sensitivity for blood pressure, wouldn't it have been a good idea to treat a control group with a diet rich in salt only without L-NAME treatment in order to verify how the intake of excess salt contributes to the increase in blood pressure and in this case whether Arcrigenin can reduce systolic blood pressure?

-In the paragraph 3.2 and 3.3 the figure numbers reported are incorrect

-In the Figures 3C-E and 4E could the authors clarify whether aortic vascular smooth muscle cells are represented or is it a section of the mice thoracic aortas?

-In each western blotting figure the samples name could be reported to show clearer results

-In the figure 4.1 is reported TMEE instead of TMEM16

Author Response

Thank you for your time and patience. We sincerely appreciate the thoughtful comments and constructive suggestions, which have helped us to substantially improve the quality of this manuscript. To facilitate your review, we have marked the reviewer’s comments in bold, and our new data and revisions are indicated in red. Although we tried our best to provide plausible explanations and incorporate all feedback, we would also like to hear from you if further improvement is warranted. We look forward to publishing our article in Biomedicines.

Point 1: In the abstract there is a small mistake: the abbreviation EMS1 instead of ESM1

Response 1: We are so sorry for the mistake, the abbreviation ESM1 can be seen in the revised abstract section.

Point 2: Since the normotensive population show salt sensitivity for blood pressure, wouldn't it have been a good idea to treat a control group with a diet rich in salt only without L-NAME treatment in order to verify how the intake of excess salt contributes to the increase in blood pressure and in this case whether Arcrigenin can reduce systolic blood pressure?

Response 2: Thank you for your insightful comments. Individuals show different salt sensitivity for BP, and studies showed that nearly 50% of the hypertensive and 25% of the normotensive population are affected. The reason why we chose mice with normal diet as the control group was to illustrate that LNAME can induce salt-sensitive hypertension mouse model. We don’t think it is a good idea to treat a control group with a diet rich in salt only without L-NAME treatment, there may be no difference of systolic blood pressure between the two group in this case. But we still believe arcrigenin can reduce systolic blood pressure in this case since arcrigenin can attenuate vascular inflammation.   

Point 3: In the paragraph 3.2 and 3.3 the figure numbers reported are incorrect

Response 3: We feel so sorry for the mistakes, and the correct numbers can be seen in the paragraph 3.2 and 3.3.

Point 4: In the Figures 3C-E and 4E could the authors clarify whether aortic vascular smooth muscle cells are represented or is it a section of the mice thoracic aortas?

Response 4: We appreciate this comment, it is a section of the mice thoracic aortas in Figures 3C-E and 4E. These modifications can be seen in the figure legends of Figures 3,4 and 6.

Point 5: In each western blotting figure the samples name could be reported to show clearer results

Response 5: We sincerely appreciate the comment, revised samples name can be seen in each western blotting figure.

Point 6: In the figure 4.1 is reported TMEE instead of TMEM16

Response 6: Thanks for the comment, we have revised the mistake in the Figure 4.

Reviewer 2 Report

Paper titled (Arcrigenin attenuates vascular inflammation induced by high salt through TMEM16A/ESM1/VCAM-1 pathway) studied the role of arctigenin in attenuating high saly induced inflammatory reactions and mechanism of this amelioration. I find the following revisions are mandatory:

1- Data should be presented as mean+-SD (not SE) this is as authors do not cover the universe for this study.

2- Stat analysis: what type of ANOVA was applied? why shy in mentioning the post-hoc test?

3- Authors have to check the normality of distribution of the results by a suitable post hoc test (such as Shapiro-Wilk test or K-S test) before deciding to choose certain ANOVA. If the normality test indicated normal dist of the data, so use one-way ANOVA, if not, use non parametric ANOVA

In all cases choose a suitable post-hoc test

4- Authors should give the source of chemicals, kits and antibodies completely and consistently (code, company, town, state and country) & version for software

5- Fig 2: authors should add an example of how they measured the thickness & what image analysis software used?

Author Response

Thank you for your time and patience. We sincerely appreciate the thoughtful comments and constructive suggestions, which have helped us to substantially improve the quality of this manuscript. To facilitate your review, we have marked the reviewer’s comments in bold, and our new data and revisions are indicated in red. Although we tried our best to provide plausible explanations and incorporate all feedback, we would also like to hear from you if further improvement is warranted. We look forward to publishing our article in Biomedicines.

Point 1: Data should be presented as mean+-SD (not SE) this is as authors do not cover the universe for this study.

Response 1: Thank you for your insightful comment, we are so sorry we made the mistakes, data in the study are presented as mean± SD, mean± SE was written wrongly.

Point 2: Stat analysis: what type of ANOVA was applied? why shy in mentioning the post-hoc test?

Response 2: We appreciate this comment. The results were analyzed by one -way ANOVA analysis of variance and post hoc analysis with Sidak correction. These modifications can be seen in the revised materials and methods section (2.11. Statistical Analysis, figure) .

Point 3: Authors have to check the normality of distribution of the results by a suitable post hoc test (such as Shapiro-Wilk test or K-S test) before deciding to choose certain ANOVA. If the normality test indicated normal dist of the data, so use one-way ANOVA, if not, use non parametric ANOVA. In all cases choose a suitable post-hoc test.

Response 3: Thank you for the suggestion. Kolmogorov-Smirnov test is used to check the normality of the distribution of the results. According to the type of data, the statistical significance was determined by using the Chi-square and t-test between two groups. Differences among three or more groups were determined using one-way analysis of variance (ANOVA) and non parametric ANOV tests, and post hoc analysis with Sidak correction was applied. These modifications can be seen in the revised materials and methods section (2.11. Statistical Analysis)

Point 4: Authors should give the source of chemicals, kits and antibodies completely and consistently (code, company, town, state and country) & version for software

Response 4: Thank you for your comment. All sources of chemicals, kits and antibodies were revised, these modifications can be seen in the revised materials and methods section.

Point 5: Fig 2: authors should add an example of how they measured the thickness & what image analysis software used?

Response 5: Thank you, this is a very insightful comment, we have added an example of how we measured the thickness [revised Figure 2B]. And the CaseViewer, 3DHISTECH, version 2.4.0 was used to perform image analysis.

Reviewer 3 Report

This study examines the role of Arcrigenin in the inhibition of high salt responses in the cardiovascular system.

Major Comments:

1. Is the growth in the medical layer in Figure 1 the result of proliferation and migration of smooth muscle cells. Please consider including staining for smooth muscle cells and markers to inflammation in vivo in the tissue.

2. The fluorescent images in figure 3, 4, and 6 are difficult to interpret. Please include a zoomed in image and please quantify the images.

3. Please quantify western blots in Figure 4 and 6 as well to help add to the findings in the manuscript. 

4. What are the effects when TMEM16A is over-expressed in aortic smooth muscle cells? Does over-expression exacerbate the high-salt response? 

Author Response

Thank you for your time and patience. We sincerely appreciate the thoughtful comments and constructive suggestions, which have helped us to substantially improve the quality of this manuscript. To facilitate your review, we have marked the reviewer’s comments in bold, and our new data and revisions are indicated in red. Although we tried our best to provide plausible explanations and incorporate all feedback, we would also like to hear from you if further improvement is warranted. We look forward to publishing our article in Biomedicines.

Point 1:  Is the growth in the medical layer in Figure 1 the result of proliferation and migration of smooth muscle cells. Please consider including staining for smooth muscle cells and markers to inflammation in vivo in the tissue.

Response 1: Thank you for your comments. Please forgive us for misunderstanding the comment 1. Figure 1 in our study showed the flow chart of animal model. And Supplementary Figure 2 showed the adhesion assay of Calcein-AM-labled macrophages adherent for smooth muscle cells with Immunofluorescence stanining in different groups.

Point 2: The fluorescent images in figure 3, 4, and 6 are difficult to interpret. Please include a zoomed in image and please quantify the images.

Response 2: As you suggested, we pay more attention to the fluorescent images in figure 3, 4, and 6. We have zoomed in images and quantified the images, the revised figures can be seen in Figure 3,4 and 6.

Point 3: Please quantify western blots in Figure 4 and 6 as well to help add to the findings in the manuscript. 

Response 3: Thank you for your insightful comments. We have quantified the images, the revised figures can be seen in Figure 4 and 6 in the manuscript.

Point 4: What are the effects when TMEM16A is over-expressed in aortic smooth muscle cells? Does over-expression exacerbate the high-salt response?

Response 4: Thank you for your comment. Firstly, another study of our team that is going to be published proved TMEM16A abundantly expressed in mesenteric resistance arteries of salt-sensitive rates and possibly played a key role in the changes of vascular function induced by high sodium intake. And in the study, we observed TMEM16A affects the recruitment of inflammatory cells and lead the abnormal vascular inflammation. Secondly, high-salt can lead overexpression of TMEM16A. So far, we don’t understand whether over-expression of TMEM16A exacerbate the high-salt response, and there is little study to focus on it. Thank you for your insightful comment again, we will explore further relationships between TMEM16A and high salt in the follow-up work.

Round 2

Reviewer 1 Report

I have no further comments

Author Response

Thank you for your comment.

Reviewer 2 Report

Why the role of LNAME Is not mentioned in Title & aim?

Author Response

Thank you for your time and patience. We sincerely appreciate the thoughtful comments and constructive suggestions. To facilitate your review, we have marked the reviewer’s comments in bold, and our response is indicated in red. Although we tried our best to provide plausible explanations and incorporate all feedback, we would also like to hear from you if further improvement is warranted. We look forward to publishing our article in Biomedicines.

Point: Why the role of LNAME is not mentioned in Title & aim?

Response: Thank you for your insightful comment. Firstly, to emphasize the role that TMEM16A can attenuate vascular inflammation in vivo and in vitro, we do not mention the role of L-NAME in title & aim. And L-NAME has been used to induce increased blood pressure for about 20 years, which is a mature and well-known model of hypertension, so we do not introduce too much about it. If it is necessary to add L-NAME to the title and aim, please feel free to let us know.

Reviewer 3 Report

No additional comments. 

Author Response

Thank you for your comment.